# Schisandrin B Attenuates Hepatic Stellate Cell Activation and Promotes Apoptosis to Protect against Liver Fibrosis

**DOI:** 10.3390/molecules26226882

**Published:** 2021-11-15

**Authors:** Zhiman Li, Lijuan Zhao, Yunshi Xia, Jianbo Chen, Mei Hua, Yinshi Sun

**Affiliations:** 1Institute of Special Animal and Plant Sciences, Chinese Academy of Agricultural Sciences, Changchun 130112, China; lzm091215@163.com (Z.L.); zlj3085@163.com (L.Z.); chenjianbo00882@126.com (J.C.); huamei@caas.cn (M.H.); 2College of Chinese Medicinal Materials, Jilin Agricultural University, Changchun 130118, China; 13947598870@163.com

**Keywords:** schisandrin B, hepatic stellate cell, apoptosis, TGF-β1

## Abstract

The activation of hepatic stellate cells (HSC) plays a key role in the progression of hepatic fibrosis, it is essential to remove activated HSC through apoptosis to reverse hepatic fibrosis. Schisandrin B (Sch B) is the main chemical component of schisandrin lignan, and it has been reported to have good hepatoprotective effects. However, Schisandrin B on HSC apoptosis remains unclear. In our study, we stimulated the HSC-T6 and LX-2 cell lines with TGF-β1 to induce cell activation, and the proliferation and apoptosis of the activated HSC-T6 and LX-2 cells were detected after treatment with different doses of Schisandrin B. Flow cytometry results showed that Sch B significantly reduced the activity of activated HSC-T6 and LX-2 cells and significantly induced apoptosis. In addition, the cleaved-Caspase-3 levels were increased, the Bax activity was increased, and the Bcl-2 expression was decreased in HSC-T6 and LX-2 cells treated with Sch B. Our study showed that Sch B inhibited the TGF-β1-induced activity of hepatic stellate cells by promoting apoptosis.

## 1. Introduction

Liver fibrosis is caused by a variety of pathological factors associated with intrahepatic connective tissue dysplasia, which results in the pathological process of intrahepatic diffuse extracellular matrix deposition [1,2]. Hepatic fibrosis is caused by many chronic injuries, which is involved in most of the chronic liver disease, such as viral hepatitis, cirrhosis or hepatocellular carcinoma [3]. In normal livers, hepatic stellate cells (HSCs) exist in a quiescent state and function in the storage and metabolism of vitamin A and retinoids, acting as pericytes of sinusoidal endothelial cells [4]. When exposed to various external stimuli, HSCs become activated. HSCs transform from a resting phenotype to a myofibroblast phenotype. Myofibroblasts produce extracellular matrix (ECM) and express increased levels of α-smooth muscle actin (α-SMA), platelet-derived growth factor (PDGF), type I Collagen (Collagen I), and tissue inhibitor of metalloproteinase 1 (TIMP1) in response to various cytokines, chemokines and growth factors [5,6,7,8]. In the pathological progression of hepatic fibrosis, a large number of activated HSCs are thought to undergo one of two processes: one process is the transition from an “activated type” to a “static type”, and the other process is apoptosis [9]. Drugs designed to inhibit HSC activation, induce cell apoptosis and prevent ECM deposition have achieved certain effects in the treatment of experimental liver fibrosis. Therefore, activated HSC is an important marker for the development of liver fibrosis, and many attempts have been made to identify drugs that reduce the number of activated proliferating HSCs by inducing HSC apoptosis; this strategy can be considered a feasible approach the treatment of fibrosis.

Schisandra is a dry ripe fruit of medicinal vine plant *Schisandra chinensis* (Turcz.) Baill. Schisandra is a known raw material that can be used in health food. Health food containing schisandra mainly includes improving sleep, auxiliary protection against chemical liver injury, enhancing immunity, relieving physical fatigue and anti-aging, all of which are consistent with the traditional benefits of schisandra [10]. Schisandra B (Sch B) is a lignan compound isolated from *Schisandra chinensis* (Turcz) Baill [11,12]. Sch B has anti-inflammatory, liver-protecting, anti-renal toxicity and anti-cancer activities [13,14,15,16,17]. Sch B inhibits the proliferation of human gastric cancer by targeting the cell cycle [18]. Sch B inhibited the secretion of IL-1β, TNF-α, IL-6 and HMGB1 in LPS-activated RAW264.7 cells [19]. Sch B alleviates enteritis by inhibiting TH17 cell differentiation [13]. Although it has been reported that Schisandra chinensis exerts protective effects against liver damage, the actual bioactive molecules have not been well identified. It is not clear which hepatocytes Sch B affects to protect against liver injury. Therefore, the aim of this study was to evaluate the potential of using Sch B to induce HSC apoptosis to protect against liver fibrosis.

## 2. Results

### 2.1. Sch B Inhibits the Proliferation of Activated Hepatic Stellate Cells

To confirm that Sch B inhibited HSC-T6 and LX-2 activation, a CCK-8 assay was used to detect the effects of 24-h treatment with 0–200 μM Sch B on activated HSC-T6 and LX-2 cells (Figure 1). Compared to the control treatment, Sch B at concentrations ranging from 1.56 to 200 μM significantly reduced the activity of HSC-T6 and LX-2 cells (*p* < 0.01, *p* < 0.001). In addition, SCh B at a concentration of 25 μM reduced HSC-T6 cell viability down to 53.65% (Figure 1A). SCh B at a concentration of 25 μM decreased the viability of LX-2 cells down to 52.7% (Figure 1B). The calculated LD50 of HSC-T6 cells and LX-2 cells was 40.615 µM and 46.65 µM Sch B, respectively. Therefore, 1.56, 3.125, 6.25, 12.5 and 25 μM Sch B were selected for subsequent studies with HSC-T6 and LX-2 cells. These results suggest that Sch B inhibits the proliferation of activated HSC-T6 and LX-2 in a dose-dependent manner.

### 2.2. Sch B Attenuated the Secretion of Profibrotic Proteins by TGF-β1-Activated HSCs

TGF-β1 is one of the most effective fibrogenic factors and can regulate cell differentiation, migration and extracellular matrix (ECM) component synthesis. We treated HSC-T6 and LX-2 cells with TGF-β1 to generate activated HSCs (Figure 2A). Sch B treatment exerted the opposite effect on protein secretion compared with TGF-β1 treatment. We identified three differentially secreted proteins by adding Sch B to activated HSCs. As shown in Figure 2B, compared with TGF-β1-treated HSC-T6 cells, HSC-T6 cells treated with TGF-β1 and Sch B at a concentration of 3.125–25 μM exhibited significantly decreased α-SMA and Collagen I protein expression (*p* < 0.01, *p* < 0.001). Sch B at a concentration of 6.25–25 μM significantly inhibited TIMP1 protein expression in a dose-dependent manner (*p* < 0.001). As shown in Figure 2C, compared with the TGF-β1-treated group, the group treated with Sch B at concentrations of 3.125 and 25 μM exhibited significantly decreased α-SMA protein expression (*p* < 0.05, *p* < 0.001). Sch B at 6.25 to 25 μM significantly inhibited the level of Collagen I (*p* < 0.01, *p* < 0.001). As the concentration ranged from 6.25 to 25 μM, Sch B decreased TIMP1 protein expression. These results indicated that Sch B could effectively inhibit the secretion of proteins associated with hepatic fibrosis.

### 2.3. Sch B Suppressed α-SMA and Collagen I Protein Expression by TGF-β1-Activated HSCs

As mentioned in Figure 3, immunofluorescence analysis showed that α-SMA and Collagen I were highly expressed in activated HSC-T6 cells. Compared with TGF-β1-treated HSC cells, Sch B-treated cells exhibited decreased expression levels of α-SMA and Collagen I in a dose-dependent manner (*p* < 0.05, *p* < 0.001) in Figure 3A. Similarly, in activated LX-2 cells, different doses of Sch B significantly reduced Collagen I protein expression in a dose-dependent manner. The activated LX-2 protein expression of α-SMA was significantly decreased after treatment with 3.125, 12.5 and 25 μM Sch B and reached its lowest level after treatment with 25 μM Sch B (*p* < 0.05, *p* < 0.001) in Figure 3B. All the results described above indicate that Sch B could effectively inhibit the activation of HSC-T6 and LX-2 cells.

### 2.4. Sch B Causes Progressive HSCs Apoptosis

To investigate whether the inhibitory effect of Sch B on HSC viability was related to apoptosis, we used flow cytometry to detect the percentage of apoptotic cells, and the analysis included both early and late apoptotic cells. The results showed that compared with TGF-β1 treatment, Sch B treatment significantly increased the Annexin V/PI staining rate of activated HSC-T6 cells in a dose-dependent manner (*p* < 0.001, Figure 4A). The Annexin V/PI staining rate of activated LX-2 cells treated with 25 and 50 μM Sch B was significantly higher than that of TGF-β1-treated cells and significantly increased in a dose-dependent manner (*p* < 0.001, Figure 4B).

### 2.5. Sch B Promotes Apoptotic Factor Expression of Activated HSCs

As shown in Figure 5, compared with TGF-β1, Sch B at concentrations of 6.25, 12.5, and 25 µM significantly increased cleaved-Cas3 expression in HSC-T6 and LX-2 cells (*p* < 0.01, *p* < 0.001). Bcl-2 expression was significantly decreased in HSC-T6 cells after 24 h of treatment with 12.5 and 25 µM Sch B (*p* < 0.01, *p* < 0.001). Sch B at concentrations from 3.125 to 25 µM significantly inhibited Bcl-2 expression in LX-2 cells (*p* < 0.01, *p* < 0.001). Compared with TGF-β1 treatment, 24 h of treatment with Sch B significantly increased the expression of Bax at doses of 1.56 μM to 25 μM in HSC-T6 cells and at doses of 3.125 μM to 25 μM in LX-2 cells. The results showed that Sch B could promote apoptosis in a dose-dependent manner.

## 3. Discussion

Liver fibrosis and cirrhosis are the results of most chronic liver injuries and present difficult clinical challenges. The occurrence and reversal of hepatic fibrosis mainly involves the processes of the generation, accumulation and degradation of cytoplasmic matrix. Activation of HSCs is driven by a variety of mediators, such as reactive oxygen species, chemokines, growth factors, matrix hardness, stromal cell proteins, and injury-related molecular patterns, which are also secreted by adjacent cells and signal HSC scar formation in an autosecretory and/or paracrine manner [20,21]. TGF-β1 plays an important role in the occurrence and development of hepatic fibrosis and is considered the most important fibrogenic factor [22].

Resting HSC degranulation releases vitamin A and morphologically changes HSCs into an activated myofibroblast state; these activated myofibroblasts significantly increase their expression of α-SMA, collagen I and TIMP1, leading to changes in the amount and composition of ECM and eventually causing excessive insoluble matrix accumulation [23]. In our current work, TGF-β1, a positive regulator of ECM production that functions in an autocrine or paracrine manner, could induce HSC activation and proliferation. TGF-β1 was used to induce T-HSC and LX-2 cells to proliferate, while Sch B inhibited the viability of activated T-HSC and LX-2 cells (Figure 1). Western blotting together with immunofluorescence analysis revealed that α-SMA and Collagen I are mainly produced by activated HSCs. Sch B reduced the levels of the fibrosis markers α-SMA and Collagen I in activated HSC-T6 and LX-2 cells (Figure 2 and Figure 3), balanced extracellular matrix synthesis, and inhibited hepatic stellate cell activation to protect against liver fibrosis.

The imbalance in the ECM in the liver leads to liver fibrosis, TIMP1 is a key enzyme that regulates ECM metabolism, and HSCs are the main source of these proteins in the liver [24]. We examined the effect of Sch B on the expression of these enzymes that are critical for ECM metabolism. Sch B downregulated the protein expression of TIMP1 in HSC-T6 and LX-2 cells in a dose-dependent manner (Figure 2).

The central link in the mechanism underlying liver fibrosis development is the activation of HSCs, which results in abnormal deposition of ECM; the number of activated HSCs is decreased through apoptosis to prevent or reverse the occurrence of liver fibrosis [25]. Flow cytometry showed that the annexin V/PI staining rate of Sch B-treated HSC-T6 and LX-2 cells was significantly higher than that of TGF-β1-treated control cells in a dose-dependent manner (Figure 4). Mitochondria, as energy metabolism centers of eukaryotic cells, are involved in cell transmission and apoptosis [26]. In the mitochondria, Bcl-2 and Bax can form ion channels, and they may participate in regulating some phenomena related to cell apoptosis, such as changes in mitochondrial permeability (large hole formation) and mitochondrial release of apoptosis protein activated factor and apoptosis inducing factor (AIF). Excessive expression of Bcl-2 inhibits changes in mitochondrial permeability and affects the formation of large holes, thus inhibiting apoptosis [27]. After this process, proapoptotic molecules released from the mitochondria help to activate members of the caspase family and other catabolic enzymes [28]. The results showed that the cleaved fragment of Cas-3 was activated in the Sch B-treated group, indicating that the activation of Cas-3 in HSC-T6 and LX-2 cells was significantly increased in the Sch B-treated group. The level of the antiapoptotic factor Bcl-2 was decreased, and the expression of the proapoptotic factor Bax was increased, indicating that Sch B can promote the apoptosis of activated HSC-T6 and LX-2 cells. During the process of promoting HSC-T6 and LX-2 cell apoptosis, Sch B significantly enhanced cell apoptosis and caspase activity, which may be the cause of mitochondrial function loss.

In conclusion, our study provides preliminary evidence that Sch B inhibits TGF-β1-stimulated HSC activation by inhibiting HSC-T6 and LX-2 cell proliferation and promoting apoptosis, thus obstructing the process of liver fibrosis.

## 4. Materials and Methods

### 4.1. Reagents

The 98% purity of Schisandrin B was purchased from Shanghai Yuanye Biotechnology Co., Ltd (Shanghai, China). The Rat HSC line HSC-T6 and the human HSC line LX-2 cells were obtained from Procell Life Science and Technology Co.,Ltd (Wuhan, China). DMSO was purchased from Sigma-Aldrich (Shanghai, China) Trading Co, Ltd. DMEM, FBS and Penicillin Streptomycin were from Gibco/Invitrogen (Grand Island, NY, USA). Cell Counting Kit-8 were obtained from DOJINDO (Kumamoto Prefecture, Kyushu Island, Japan). TGF-β1 was purchased from Peprotech (Rocky Hill, NJ, USA). Anti-Caspase-3 (sc7272) antibody were purchased from Santa Cruz Biotechnology Inc (Santa Cruz, CA, USA). The anti-α-SMA (ab5694), anti-Collagen I (ab34710), anti-TIMP1 (ab61224), anti-BCL-2 (ab182858), anti-Bax (ab32563) anti-GAPDH (ab8245) antibody, horseradish-peroxidase (HRP) conjugated secondary antibody (a11008) and Fluoroshield Mounting Medium with DAPI were purchased from Abcam (Cambridge, MA, USA). The BCA Protein Assay Kit and BeyoECL plus kit was obtained from Beyotime (Shanghai, China). Goat anti-Rabbit IgG (H + L) Cross-Adsorbed Secondary Antibody, Alexa Fluor 488 were obtained from ThermoFisher Scientific (Waltham, MA, USA). FITC/PI cell apoptosis kit were purchased from Keygen Biotechnology Co., Ltd (Nanjing, China).

### 4.2. Preparation of Sch B Solutions 

Sch B dissolved DMSO and prepared it into 50 mM stock solution, which was completely dissolved after standing in a warm water bath at 37 °C for 5 min. The cell activity assay was performed using a gradient dilution method with 0–200 µM Sch B.

### 4.3. Cell Culture 

The rat HSC line HSC-T6 was cultured in Dulbecco’s modified Eagle’s medium (DMEM), and the human HSC line LX-2 was cultured in 1640 DMEM supplemented with 10% fetal bovine serum (FBS), 100 U/mL penicillin and 100 U/mL streptomycin in a 37 °C, 5% CO_2_ saturated humidity environment. For the experiment, HSC-T6 and LX-2 were plated in 6-well plates at a density of 5 × 105/well and were exposed to TGF-β1 (10 ng/mL) for 2 h. The cultured wells containing TGF-β1 were added with different doses of Sch B for 22 h.

### 4.4. CCK-8 Determination

HSC-T6 and LX-2 cells (1 × 10^4^) were seeded in a 96-well plate and incubated overnight, and then, the medium was changed to medium supplemented with 10 ng/mL TGF-β1. After stimulation for 2 h, different concentrations of Sch B (0–200 μM) were added. Then, the cells were incubated for 24 h, and 10 μL of Cell Counting Kit-8 (CCK-8) assay reagent was added to each well. The optical density (OD) value was measured at 490 nm using a spectrophotometer.

### 4.5. Western Blotblotting

The total proteins were harvested from the cells with RIPA buffer and analyzed by Western blotting analysis. The protein concentrations were determined using a BCA assay. SDS-PAGE (10–15%) gels were used to separate equal amounts of the protein samples, and the proteins were transferred onto PVDF membranes, which were then blocked for 1 h. The blots were incubated with diluted primary antibodies, including antibodies against α-SMA, Collagen I, TIMP1, Caspase-3, Bcl-2, and GAPDH, at 4 °C overnight, followed by detection using an HRP-conjugated secondary antibody (1 h, room temperature). The protein bands were visualized by the BeyoECL plus kit.

### 4.6. Flow Cytometry Analysis

HSC-T6 and LX-2 cells were digested with EDTA-free trypsin and then stained according to the instructions of the FITC/PI apoptosis kit. The stained cells were detected by flow cytometry after 5–15 min.

### 4.7. Immunofluorescence Staining

After treatment, the HSC-T6 cells or LX-2 cells were fixed in 4% paraformaldehyde at room temperature for 20 min. The cells were then washed and permeabilized with PBS containing 0.1% Triton X-100 and blocked with 5% goat serum in PBS. Anti-α-SMA or anti-Collagen I was added to the cells. After incubation at 4 °C overnight, the cells were washed 3 times with PBS. Fluorescence secondary antibody was incubated at room temperature for 1 h. After adding DAPI, images were obtained under a microscope.

### 4.8. Statistical Analysis

All the data are expressed as the mean ± SD. One-way ANOVA and Tukey’s multiple comparison tests were used for data analysis. GraphPad Prism version 5 (GraphPad Software, San Diego, CA, USA) for statistical analysis of data. A *p* value of <0.001 was considered statistically significant.

## Figures and Tables

**Figure 1 molecules-26-06882-f001:**
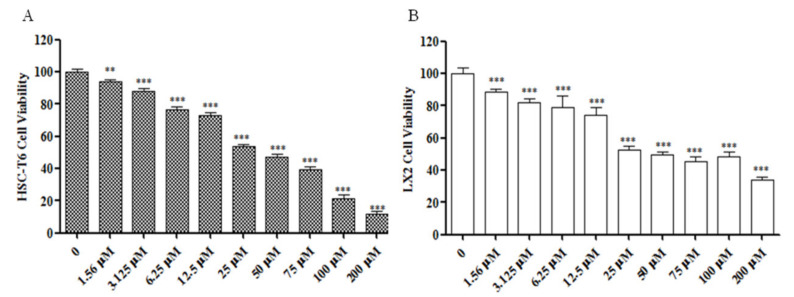
Sch B inhibited the viability of activited HSC-T6 and LX-2 cells. (**A**) Effect of Sch B on the viability of HSC-T6 cells shown by CCK8 assay. (**B**) Effect of Sch B on the viability of LX-2 cells shown by CCK8 assay. The results are presented as means ± SD, ** *p* < 0.01, *** *p* < 0.001 compared with the TGF-β1 group.

**Figure 2 molecules-26-06882-f002:**
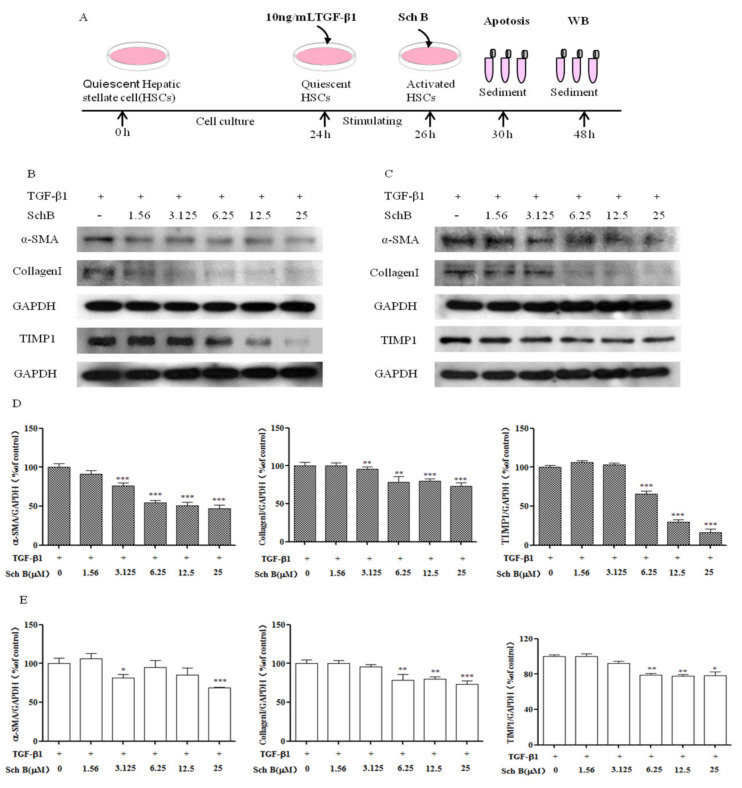
Identification of the Sch B anti-fibrotic effect in vitro. (**A**) Schematic representation of the experimental schedule. Alterations of Sch B on the expression of profibrotic proteins levels of α-SMA, TIMP-1, and Collagen I were determined by western blotting. Sch B suppressed the expressions of α-SMA, Collagen I and TIMP-1 in a dose-dependent manner in (**B**) HSC-T6 cells and (**C**) LX-2 cells. (**D**) Densitometric analysis of α-SMA, TIMP-1, and Collagen I of HSC-T6 cells. (**E**) Densitometric analysis of α-SMA, TIMP-1, and Collagen I of LX-2 cells. The values are expressed as the mean ± SD. * *p* < 0.05, ** *p* < 0.01, *** *p* < 0.001 compared with the TGF-β1 group.

**Figure 3 molecules-26-06882-f003:**
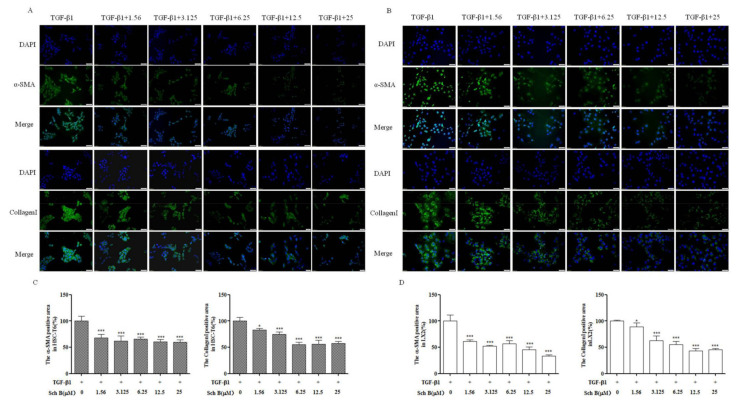
Sch B suppressed the proteins expression of α-SMA and Collagen I. Immunofluorescence analysis of α-SMA and Collagen I expression in HSC-T6 (**A**) and LX-2 (**B**) cells; magnification, 200×. The positive areas were quantified in HSC-T6 (**C**) and LX-2 (**D**) cells. * *p* < 0.05, *** *p* < 0.001, significantly different when compared with the TGF-β1-activated cells.

**Figure 4 molecules-26-06882-f004:**
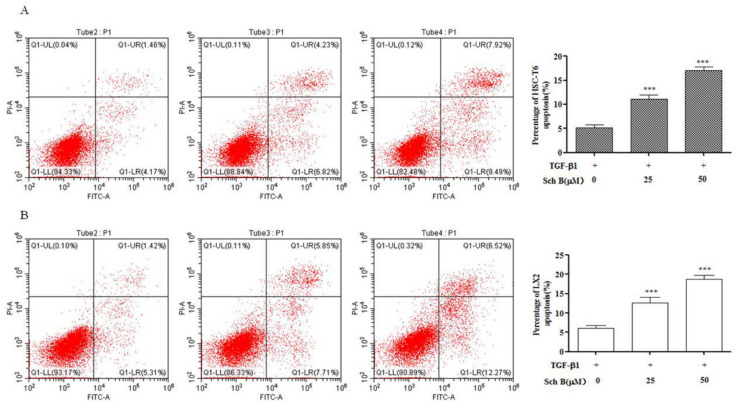
Sch B induced HSCs apoptosis. (**A**) HSC-T6 cells were treated with 0, 25 or 50 μM Sch B for 24 h, (**B**) LX-2 cells were treated with 0, 25 or 50 μM Sch B for 24 h and apoptosis was assessed by flow cytometry. The results shown are means ± SD, *** *p* < 0.001 compared with the TGF-β1 group.

**Figure 5 molecules-26-06882-f005:**
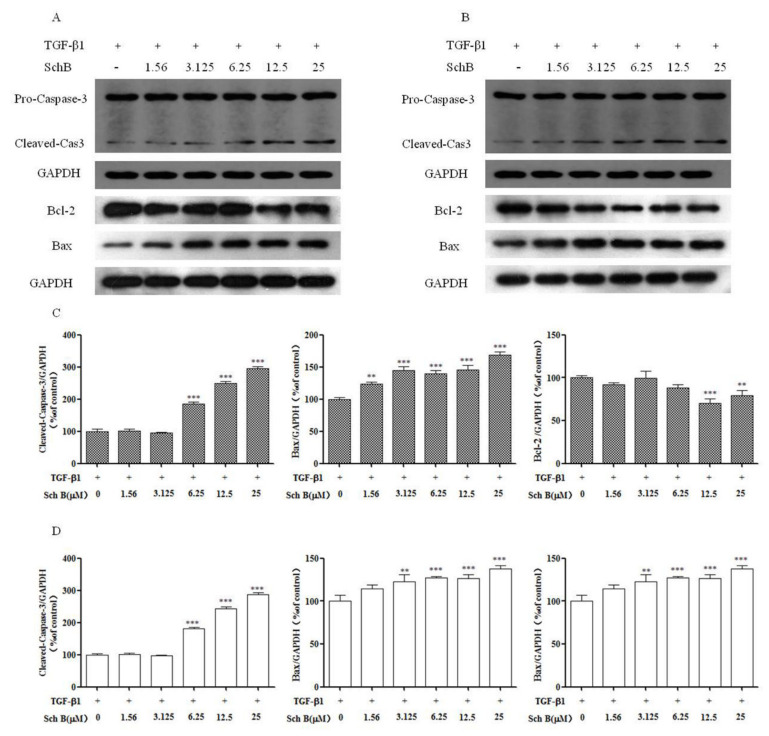
Effects of Sch B on hepatic stellate cells apoptosis factors. The expression of Bcl-2, Bax, and Caspase-3 of HSC-T6 (**A**) and LX-2 (**B**) were determined by western blots. (**C**) Densitometric analysis of cleaved-Cas3, Bax and Bcl-2 of HSC-T6 cells. (**D**) Densitometric analysis of cleaved-Cas3, Bax and Bcl-2 of LX-2 cells. The protein density of bands in western blots were detected using Bio-Rad Quantity One software. Data are presented as means ± SD, ** *p* < 0.01, *** *p* < 0.001 compared with the TGF-β1 group.

## Data Availability

Not applicable.

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
