# Peer review of "Schisandrin B Attenuates Hepatic Stellate Cell Activation and Promotes Apoptosis to Protect against Liver Fibrosis"

_molecules, 2021, doi:10.3390/molecules26226882_

Round 1

Reviewer 1 Report

The authors did perform all the necessary improvements and modifications to the presentation of their manuscript so I suggest the manuscript should be accepted for publication in the revised form.

Author Response

Thank you for your hard work on this manuscript. Your serious and professional scientific research and writing attitude is worth my learning. Your comments are all valuable and very helpful for revising and improving our paper. Thanks again. 

Reviewer 2 Report

Authors have covered all my points. Just a couple of minor corrections

Lane 92, after the following sentence : "HSC-T6 and LX-2 cells with TGF-β1 to generate activated HSCs"   add (Figure 2A)

Lane 95   Change "Figure 2A"   by "Figure 2B"

Author Response

Thank you for your hard work on this manuscript. Your serious and professional scientific research and writing attitude is worth my learning. Your comments are all valuable and very helpful for revising and improving our paper.

1.Lane 92, after the following sentence : "HSC-T6 and LX-2 cells with TGF-β1 to generate activated HSCs"   add (Figure 2A)

Lane 95   Change "Figure 2A"   by "Figure 2B"

Response:I have added “(Figure 2A)” in lane 92, changed "Figure 2A" by "Figure 2B" and  "Figure 2B" by "Figure 2C".

This manuscript is a resubmission of an earlier submission. The following is a list of the peer review reports and author responses from that submission.

Round 1

Reviewer 1 Report

The manuscript investigates the inhibitory efficacy of Schisandrin B on rat and human HSC cells activation. The authors also investigate apoptosis activities by expressions of caspase 3, Bcl-2 and Bax. Overall the manuscript is well written in terms of language and understandability with only minor proposed revision.

  1. Figure 2 (B) was poor resolution for alpha-SMA and Collagen I.
  2. Moreover, Figure 2(B) TIMP1 was found to be a dose-depending decreasing  fashion due to variant GAPDH not directly by TIMP1 expressions. Please elaborate more.
  3. Figure 4 (D) Bcl-2/GAPDH was not shown but instead Bax/GAPDH exhibited twice. 
  4. Please provide the Schisandrin B source or a brief manufacturing process.

Author Response

1.Figure 2 (B) was poor resolution for alpha-SMA and Collagen I.

Response: Special thanks to you for your good comments, alpha-SMA and Collagen I were replaced according to the Reviewer’s comments.

2.Moreover, Figure 2(B) TIMP1 was found to be a dose-depending decreasing fashion due to variant GAPDH not directly by TIMP1 expressions. Please elaborate more.

Response: Thanks for the reviewers’ attention to our manuscript. We are very sorry that due to my carelessness description, then replaced TIMP1 and GAPDH with a more distinct one, and again analyzed the density values. The concentration ranged from 6.25 to 25 μM, Sch B decreased TIMP1 protein expression.

3.Figure 4 (D) Bcl-2/GAPDH was not shown but instead Bax/GAPDH exhibited twice.

Response: We are grateful to the reviewer for pointing out our error. Bcl-2/GAPDH was added in Figure 4 (D).

4.Please provide the Schisandrin B source or a brief manufacturing process.

Response: Thanks for the reviewers’ attention to our manuscript, the 98% purity of Schisandrin B was purchased from Shanghai Yuanye Bio-Technology Co., Ltd (Shanghai, China).

Reviewer 2 Report

In this study, the authors investigated the effects of Schisandrin B on the induction of apoptosis of activated hepatic stellate cells that play a key role in the progression of hepatic fibrosis. Thus, HSC-T6 and LX-2 cell lines were activated with TGF-β1, and the proliferation and apoptosis were assessed after treatment with different concentrations of Schisandrin B. The main findi gs, were the increase in cleaved-Caspase-3 levels, and Bax activity, as wewll as the decrease in Bcl-2 expression. 

The manuscript can be accepted for publication after minor modifications.

  1. The whole manuscript should be extensively checked and rephrased for plagiarism
  2. At the Results section 2.1. please rephrase: reduce/decrease viability down to  
  3. Legend of Figure 2 change Regulations to alterations
  4. Legend of Figure 4 add descriptions for C & D

Author Response

1.The whole manuscript should be extensively checked and rephrased for plagiarism

Response: We are grateful to the reviewer for professional advice, according to the editor's checked report, we will seriously check and rephrase for this manuscript.

2.At the Results section 2.1. please rephrase: reduce/decrease viability down to  

Response: Special thanks to you for your good comments, “SCh B at a concentration of 1.56~25 μM reduced HSC-T6 cell viability down to 53.65% (Figure 1B). SCh B at a concentration of 1.56~25 μM decreased the viability of LX-2 cells down to 52.7% (Figure 1C).”

3.Legend of Figure 2 change Regulations to alterations

Response: Thanks for the reviewers’ attention to our manuscript, we have changed “Regulations” to “Alterations” in Figure 2.

4.Legend of Figure 4 add descriptions for C & D

Response: We are grateful to the reviewer for professional advice, we have added Legend of Figure 4 add descriptions for C & D. “(C) The bars of cleaved-Cas3, Bax and Bcl-2 of HSC-T6 cells. (D) The bars of cleaved-Cas3, Bax and Bcl-2 of LX-2 cells.”

Reviewer 3 Report

Li and co-workers evaluate the use of Schisandrin B (Sch B), an active dibenzooctadiene lignan compound, to induce apoptosis in Hepatic Stellate Cells (HSC) to protect against liver fibrosis. The authors used an ex-vivo cellular model to show that in TGF-β1 activated rat HSC-T6 and human LX-2 cell lines, the treatment with Sch B induced apoptosis along with the reduced protein expression of pro-fibrotic proteins. Although the research topic in the manuscript is relevant for the field, this study is not well-presented and lacks organization. Experimental procedures are poorly described and, several experiments missed critical controls or conditions to support author conclusions.

Major comments.

  1. Overall, the manuscript lacks coherence and organization. For instance, the figure corresponding with section 2.3 ( lanes 98-107) is not mentioned in the text. Then this figure ( Figure 5) is presented at the end of the result section as an orphan figure. In Figure 1A, the description of the experimental schedule should be moved in Figure 2 since it describes the experiments with TGF-β1 and Sch B. In this figure, the times “30h” and “48h” did not correspond to what is described in the manuscript (Sch B treatment for 24h). The dose used for the TGF-β1 treatment needs to be included in the figure legends.
  2. The experimental procedures and statistics are poorly described, to the extent that there is not the minimum information necessary for the reader to validate or repeat the experiments. The number of experimental repetitions and biological replicas is not mentioned. Description of the antibodies used for the western blot and immunofluorescence analysis are not described. Did the antibodies used in this study recognize cleanly both rat and human proteins? Proper controls are missing or not well described in most of the experiments. Is the Sch B dissolved in buffer, water, or solvent (i.e., DMSO)? To appreciate and confirm the activation effect of the TGF-β1in HSCs, authors should show the basal protein levels or conditions for untreated cells in Figures 2, 3, 4, and 5. Are the cells washed to remove TGF-Β1 before the Sch B treatment? How does Sch B treatment alone impact apoptotic and ECM markers? Contrary to what the authors mentioned on lane 1250, there is no experiment in this work showing that the treatment alone of TGF-Β1 induces the proliferation of HSC cell lines. 
  3. The authors concluded that Sch B inhibits TGF-β1-stimulated HSC fibrosis by inhibiting cell proliferation and promoting apoptosis based on their ex vivo experiments in two cell lines. The authors need to validate these observations using an in vivo animal model to reach that conclusion.
  4. A more detailed dose and time cell viability assessment need to be included in both cell lines for the Sch B treatment ( i.e., 48 and 72h). What is the LD50 for both cell lines? Several reports showed that in other cells lines, SchB induced cell cycle arrest. The authors should determine whether Sch B-induced cytotoxicity was caused by cell cycle arrest by FACS and evaluate some markers like p21/CDKN1A and cyclin D1 levels. Another potential maker to consider is the tumor suppressor p53, which is well known to regulate both cell cycle arrest and apoptosis. The authors should consider assessing by western blot if Sch B treatment alone induces p53 or its activation by phosphorylation and how the p53 levels are impacted with Sch B treatment in TGF-β1 activated HSC cell lines.
  5. Using a Mitochondrial membrane potential (ΔΨm) assay, the authors should evaluate if Sch B treatment induces a mitochondrial function loss as part of the molecular mechanisms involved in the apoptosis induction.
  6. After the cell viability assays (Figure 1), the authors mentioned that the range of Sch B concentrations between 1.56 and 25 μM were selected for subsequent studies with HSC-T6 and LX-2 cells. Then what was the basis for testing 50 uM of concentration in the Annexin-V/ PI assay? Notably, in TGF-β1 activated LX2 cells, the apoptosis induced by Sch B at 50 uM was only ~20% (Figure 3B), while the same Sch B treatment alone, as described in Figure 1C, resulted in a 60% reduction in the cell viability. The authors should explain these differences.

Minor comments

  1. Indicated that results described in lanes 82-89 are in the LX-2 cell line.
  2. The densitometric analysis for TIMP1 western blot in LX-2 cells (Figure 2D) did not correspond to what was shown in the immunoblot image (Figure 2B), where the reduction of TIMP1 levels after SchB treatment is not appreciated.
  3. Several publications have already described the effects of Sch B on cell proliferation and apoptosis induction using distinct mammalian cell lines. The authors should discuss the relevance of their results accordingly.

Author Response

Major comments.

1.Overall, the manuscript lacks coherence and organization. For instance, the figure corresponding with section 2.3 ( lanes 98-107) is not mentioned in the text. Then this figure ( Figure 5) is presented at the end of the result section as an orphan figure. In Figure 1A, the description of the experimental schedule should be moved in Figure 2 since it describes the experiments with TGF-β1 and Sch B. In this figure, the times “30h” and “48h” did not correspond to what is described in the manuscript (Sch B treatment for 24h). The dose used for the TGF-β1 treatment needs to be included in the figure legends.

Response: Thanks for the reviewers’ attention to our manuscript, we've rearranged the graph and the result accordingly. The description of experimental schedule has been removed from Figure 1, the dose of TGF-β1 has been added to Figure Legend, and SchB treatment has been changed to 22 h in 4.3.

2.The experimental procedures and statistics are poorly described, to the extent that there is not the minimum information necessary for the reader to validate or repeat the experiments. The number of experimental repetitions and biological replicas is not mentioned. Description of the antibodies used for the western blot and immunofluorescence analysis are not described. Did the antibodies used in this study recognize cleanly both rat and human proteins? Proper controls are missing or not well described in most of the experiments. Is the Sch B dissolved in buffer, water, or solvent (i.e., DMSO)? To appreciate and confirm the activation effect of the TGF-β1in HSCs, authors should show the basal protein levels or conditions for untreated cells in Figures 2, 3, 4, and 5. Are the cells washed to remove TGF-Β1 before the Sch B treatment? How does Sch B treatment alone impact apoptotic and ECM markers? Contrary to what the authors mentioned on lane 1250, there is no experiment in this work showing that the treatment alone of TGF-Β1 induces the proliferation of HSC cell lines.

Response: Thank you very much for your professional advice. We have added reagents in 4.1, which contains the the antibody source and relevant information. In 4.2, the dissolution method of SchB and specific dosing regimen were introduced. In addition, the first group was TGF-β1 alone treatment group.

4.1Reagents

The 98% purity of Schisandrin B was purchased from Shanghai Yuanye Biotechnology Co., Ltd. The Rat HSC line HSC-T6 and the human HSC line LX-2 cells were purchased from Procell Life Science&Technology Co.,Ltd. DMSO was purchased from Sigma Biotechnology Inc. DMEM, FBS and Penicillin Streptomycin were from Gibco/Invitrogen. Cell Counting Kit-8 was purchased from Japan Tongren Company. TGF-β1 was purchased from Peprotech. Anti-Caspase-3(sc7272) antibody were purchased from Santa Cruz Biotechnology Inc. The anti-α-SMA(ab5694), anti-Collagenâ… (ab34710), anti-TIMP1(ab61224), anti-BCL-2(ab182858), anti-Bax(ab32563) anti-GAPDH(ab8245) antibody, horseradish- peroxidase (HRP) conjugated secondary antibody(a11008) and Fluoroshield Mounting Medium With DAPI were purchased from Abcam. The BCA Protein Assay Kit was obtained from Beyotime. Goat anti-Rabbit IgG (H+L) Cross-Adsorbed Secondary Antibody, Alexa Fluor 488 ThermoFisher Scientific. FITC/PI cell apoptosis kit purchased from Keygen Biotechnology Co., Ltd.

4.2. Preparation of Sch B Solutions

Sch B dissolved DMSO and prepared it into 50 mM stock solution, which was completely dissolved after standing in a warm water bath at 37℃ for 5 min. The cell activity assay was performed using a gradient dilution method with 0-200 µM Sch B.

3.The authors concluded that Sch B inhibits TGF-β1-stimulated HSC fibrosis by inhibiting cell proliferation and promoting apoptosis based on their ex vivo experiments in two cell lines. The authors need to validate these observations using an in vivo animal model to reach that conclusion.

Response:Thanks for the professional opinions, we modified the conclusion to show that “In conclusion, our study provides preliminary evidence that Sch B inhibits TGF-β1-stimulated HSC activition by inhibiting HSC-T6 and LX-2 cell proliferation and promoting apoptosis, thus obstructing the process of liver fibrosis.”

4.A more detailed dose and time cell viability assessment need to be included in both cell lines for the Sch B treatment ( i.e., 48 and 72h). What is the LD50 for both cell lines? Several reports showed that in other cells lines, SchB induced cell cycle arrest. The authors should determine whether Sch B-induced cytotoxicity was caused by cell cycle arrest by FACS and evaluate some markers like p21/CDKN1A and cyclin D1 levels. Another potential maker to consider is the tumor suppressor p53, which is well known to regulate both cell cycle arrest and apoptosis. The authors should consider assessing by western blot if Sch B treatment alone induces p53 or its activation by phosphorylation and how the p53 levels are impacted with Sch B treatment in TGF-β1 activated HSC cell lines.

Response: Thank you for your professional comments. The calculated LD50 of HSC-T6 cells and LX-2 cells was 40.615 µM and 46.65 µM respectively. In the pre-experimental experiment, SchB was used to treat the activated HSC for 22 h, and the cell apoptosis was observed under the microscope, so the viability test for 48 h or 72 h was not carried out. In another resubmission, we have evaluated whether Sch B alone induces p53 or activates p53 through phosphorylation

5.Using a Mitochondrial membrane potential (ΔΨm) assay, the authors should evaluate if Sch B treatment induces a mitochondrial function loss as part of the molecular mechanisms involved in the apoptosis induction.

Response: We are grateful to the reviewer for pointing out the following research content, We will focus on a Mitochondrial membrane potential (ΔΨm) assay.

6.After the cell viability assays (Figure 1), the authors mentioned that the range of Sch B concentrations between 1.56 and 25 μM were selected for subsequent studies with HSC-T6 and LX-2 cells. Then what was the basis for testing 50 µM of concentration in the Annexin-V/ PI assay? Notably, in TGF-β1 activated LX2 cells, the apoptosis induced by Sch B at 50 µM was only ~20% (Figure 3B), while the same Sch B treatment alone, as described in Figure 1C, resulted in a 60% reduction in the cell viability. The authors should explain these differences.

Response: Thank you very much for your comments. Your suggestion is both professional and rigorous, and I really overlooked this problem. I have repeated the CCK8 experiment on HSC-T6 and LX-2 cells. The cell viability of LX-2 was 49.37% at 50 µM (Figure 1B), and the apoptosis induced by Sch B at 50 µM was only ~20% (Figure 3B). As you mentioned, I have seriously thought that I may have lost some cells in the process of apoptosis experiment of cells.

Minor comments

1.Indicated that results described in lanes 82-89 are in the LX-2 cell line.

Response: Thank you very much for your comments. We have the results of LX-2 cell line are described.

2.The densitometric analysis for TIMP1 western blot in LX-2 cells (Figure 2D) did not correspond to what was shown in the immunoblot image (Figure 2B), where the reduction of TIMP1 levels after SchB treatment is not appreciated.

Response:Thanks for the reviewers’ attention to our manuscript. We are very sorry that due to my carelessness description, then replaced TIMP1 and GAPDH with a more distinct one, and again analyzed the density values.

3.Several publications have already described the effects of Sch B on cell proliferation and apoptosis induction using distinct mammalian cell lines. The authors should discuss the relevance of their results accordingly.

Response:Thanks for the reviewers’ attention to our manuscript. I describe the effects of Sch B on different mammalian cell lines in the introduction. “Sch B inhibits the proliferation of human gastric cancer by targeting the cell cycle [18]. Sch B inhibited the secretion of IL-1β, TNF-α, IL-6 and HMGB1 in LPS-activated RAW264.7 cells [19]. Sch B alleviates enteritis by inhibiting TH17 cell differentiation [20].”

Round 2

Reviewer 3 Report

1. Although the authors have modified some sections of the manuscript, the results' presentation is still not clear and disorganized.  The authors show carelessness in checking the revised manuscript. Most of the number figures do not match with the text description. Some sections are not clear, and moderate English changes still are required. There are several words misspelled along with the revised version, for example,
lane 40 activated  instead of "activated"
lane 91 α-MA instead of  "α-SMA"

2. There are still some critics/ questions that the authors did not answer satisfactorily. To show convincingly that the addition of TGF-ß1 is indeed activating the HSCs in their in vitro model. For that,  the authors need to show HSCT-6 and LX-2 cells basal conditions for untreated cells   (no TGF-B1, no SchB) for their western blots, immunofluorescence, and Annexin V/ PI experiments.

3. In the discussion section, lane 171,  the sentence is overstated.  The authors do not present data about  SchB inhibiting activity of TIMP1.  The figure only showed that SchB treatment reduced the protein levels of TIMP1, but not its enzymatic activity.

4. The LD50  calculated values should be included in the corresponding section of the manuscript.

5. The new paragraph in the introduction describing the source and properties for the Schisandra Chinensis is not clear ( lanes 46-50). The mention that SchB is a lignan compound is repeated twice ( lanes 50 and 57) also that SchB is isolated from Schisandra Chinensis (Turcz) Baill (lanes 44 and 51)

6. Figure legend 5 is confusing and not well described for items (C) and (D). Change "The bars" by  Densitometric analysis of cleaved.....

7. The authors still did not mention the number of repetitions for the experiments

8. In the revised PDF, the description of the experimental schedule still is in the manuscript.